# Osteopathic Treatment for Gastrointestinal Disorders in Term and Preterm Infants: A Systematic Review and Meta-Analysis

**DOI:** 10.3390/healthcare10081525

**Published:** 2022-08-12

**Authors:** Francesca Buffone, Domenico Monacis, Andrea Gianmaria Tarantino, Fulvio Dal Farra, Andrea Bergna, Massimo Agosti, Luca Vismara

**Affiliations:** 1Division of Paediatric, Manima Non-Profit Organization Social Assistance and Healthcare, 20125 Milan, Italy; 2Department of Research, SOMA—Istituto Osteopatia Milano, 20126 Milan, Italy; 3PPCR, Harvard T.H. Chan School of Public Health—ECPE, Boston, MA 02115, USA; 4Department of Humanities, Literature, Cultural Heritage, Education Sciences, University of Foggia, 71122 Foggia, Italy; 5Department of Medical Sciences and Public Health, University of Cagliari, 09124 Cagliari, Italy; 6Woman and Child Department, Varese Hospital, Insubria University, 21100 Varese, Italy; 7Division of Neurology and Neurorehabilitation, IRCCS Istituto Auxologico Italiano, 28824 Piancavallo, Italy; 8Department of Neurosciences, University of Torino, 10126 Torino, Italy

**Keywords:** osteopathic manipulative treatment, gastrointestinal function, term infant, preterm infant, newborn

## Abstract

The aim of this systematic review and meta-analysis is to evaluate the effectiveness of osteopathic manipulative treatment (OMT) for gastrointestinal disorders in term and preterm infants. Eligible studies were searched on PubMed, Scopus, Embase, Cochrane, Cinahl, and PEDro. Two reviewers independently assessed if the studies were randomized controlled trials (RCTs) and retrospective studies with OMT compared with any kind of control in term or preterm infants to improve gastrointestinal disorders. Nine articles met the eligibility criteria, investigating OMT compared with no intervention, five involving term infants, and the remaining treating preterm infants. Five studies showed low risk of bias. In the meta-analysis, two studies were included to analyze the hours of crying due to infantile colic, showing statistically significant results (ES = −2.46 [−3.05, −1.87]; *p* < 0.00001). The quality of evidence was “moderate”. The other outcomes, such as time to oral feeding, meconium excretion, weight gain, and sucking, were presented in a qualitative synthesis. OMT was substantially safe, and showed efficacy in some cases, but the conflicting evidence and lack of high-quality replication studies prevent generalization. High-quality RCTs are recommended to produce better-quality evidence.

## 1. Introduction

The gastrointestinal tract (GT) carries out different tasks such as digestive, absorptive, neuroendocrine, and immunologic functions, which are all fundamental in infant development. However, disorders in the GT can arise and threaten their health status. For instance, sucking and swallowing problems occur in preterm infants born at less than 32 to 34 weeks, requiring tube enteral feeding until the reach of the full oral feeding, which is one of the parameters required for discharge [1,2]; furthermore, sucking difficulties can occur in term infants and lead to breastfeeding problems, consequently affecting the development of the intestinal microbiota [3,4]. Other functional disorders of the GT are infantile colic, functional diarrhea, and cyclic vomiting syndrome, which occur in almost half of infants, affecting also their families and the healthcare system [5,6,7,8]; during 2014/2015 in England, the total costs were about GBP 72.3 million per year [9]. Then, looking at the therapies, the unknown pathogenesis and the nonorganic etiology make use of pharmacological interventions not fully helpful in improving the children’s health status, also considering that the use of drugs is often discouraged due to the adverse effects related to medicine assumption. Hence, the possible suggested treatments are nonpharmacological and individualized, depending on the specific problem and on the family itself [6].

Osteopathic manipulative treatment (OMT) is becoming day by day more present, showing significant benefits in the pediatric population in different studies. OMT is a noninvasive complementary therapy that uses different manipulative approaches to improve physiological response and support physical homeostasis altered by somatic dysfunction (ME93.0 in the ICD-11 coding tool [10,11,12]). Improving the child’s health through manipulation inducing better adaptation is the main objective of OMT. The treatment focuses, in particular, on the manipulation and movement of the visceral fascia. Furthermore, it has been reported that OMT has an effect on the autonomic nervous system and on the fascial system, which are two systems strictly implicated in GT development [10,13,14]. To date, there is a small number of reviews investigating the effects of OMT in the pediatric population, such as a Cochrane systematic review and meta-analysis, which showed low evidence about the significant improvements of manipulative therapies for infantile colic, a systematic review concluding that the effectiveness of OMT in pediatrics is unproven due to the paucity and the poor methodology of the included studies, and eventually a systematic scoping review, which confirmed that there is still little evidence, even though OMT can be medically tolerated when given to the low-risk profile, adding that it has a strong therapeutic benefit potential for pediatric care [15,16,17]. Notwithstanding, there is a lack of systematic reviews on OMT effects on gastrointestinal disorders and its clinical implications.

Hence, this systematic review and meta-analysis assesses whether OMT can be effective in the management of disorders of the GT in both preterm and term infants.

## 2. Materials and Methods

### 2.1. Protocol Registration

This systematic review follows the PRISMA statement, and its protocol is available on PROSPERO (https://www.crd.york.ac.uk/prospero/display_record.php?ID=CRD42021293463 Registration Number: CRD42021293463; accessed on 1 February 2022) [18].

### 2.2. Search Process

A literature search was performed to evaluate the efficacy of OMT on functional gastrointestinal disorders in newborns. Literature was searched up to June 2021 in the following databases: PubMed, Scopus, Embase, Cochrane, Cinahl, and PEDro.

On PubMed, the search strategy used was: (((((osteopathic manipulative treatment[MeSH Terms]) OR (“osteopathic manipulative treatment*”[Title])) OR (osteopath*[Title])) OR (“osteopathic manipulation*”[Title])) OR (“craniosacral therap*”[Title])) AND (((infant[MeSH Terms]) OR (infant*[Title])) OR (newborn*[Title])); on the other databases the following words were combined: “osteopathic manipulative treatment”, “osteopathic manipulation”, “craniosacral”, “gastrointestinal disorder”, “gastrointestinal function”, “colic”, “sucking”, “feeding”, “infant”, “preterm”, “premature” and “newborn”. We used the same search strategy for the other indicated databases.

### 2.3. Eligibility

The research and the screening processes were conducted according to the following inclusion criteria: randomized controlled trials (RCTs), quasi-RCTs, or retrospective studies assessing the effectiveness and/or efficacy of OMT on gastrointestinal disorder(s) or function(s) in either term or preterm infants (age at study entry < 12 months) with a control group (CG) as a comparison; furthermore, the English language was required.

Due to the intrinsic variability of OMT, no restrictions regarding time, frequency, and type of techniques (i.e., craniosacral, myofascial, visceral, soft-tissue, etc.) were applied; moreover, OMT could be alone or combined with other therapies. Instead, studies using other forms of manual therapies (e.g., chiropractic, physiotherapy) applied to the experimental group were excluded. For the CG, it could have any kind of intervention (i.e., sham, usual care, pharmacological therapy, etc.) or no intervention.

### 2.4. Study Selection and Data Collection

Rayyan QCRI software was used for records management and screening operations [19]. A reviewer checked and manually removed the possible duplicates, which were previously detected by the software. Then, two reviewers independently analyzed whether the studies fulfilled the eligibility criteria: they evaluated the title and abstract, and then examined the full text. A third reviewer contributed to resolving any disagreement. Then, two reviewers independently extracted the main characteristics of each article: first author and year of publication, study design, objectives and outcomes, sample size, sample age and percentage of male/female, type of OMT, and control intervention (dose, frequency, techniques), and main results. A third reviewer contributed to resolving any discrepancy.

### 2.5. Outcomes

The primary outcome for this review was any endpoint related to the gastrointestinal function in newborns measured at post-intervention and follow-up. Secondary outcomes involved the length of stay (LOS), parents’ care satisfaction, and possible adverse events (AEs).

### 2.6. Assessment of Risk of Bias

Two reviewers used the Cochrane Risk of Bias tool 2.0 for randomized trials (RoB2) to independently assess the methodological quality of the included RCTs [20]. This tool considers five different domains: randomization process (Criteria 1.1, 1.2, and 1.3), deviation from intended interventions (Criteria 2.1 to 2.7), missing outcome data (Criteria 3.1 to 3.4), measurement of the outcome (Criteria 4.1 to 4.5), and selection of reported results (Criteria 5.1 to 5.3). RoB2 was assessed for each domain, according to a three-point scale: low risk of bias, some concerns, and high risk of bias. In the case of disagreement, a third reviewer contributed to resolving the discrepancies.

The retrospective studies were analyzed with the Risk of Bias In Nonrandomized Studies of Interventions (ROBINS-I) [21]. It considers seven domains: bias due to confounding (Criteria 1.1 to 1.8), bias in selection of participants into the study (criteria 2.1 to 2.5), bias in classification of interventions (Criteria 3.1 to 3.3), bias due to deviations from intended interventions (Criteria 4.1 and 4.2), bias to missing data (Criteria 5.1 to 5.5), bias in measurement of outcomes (Criteria 6.1 to 6.4), and bias in selection of the reported results (criteria 7.1 to 7.3).

### 2.7. Data Synthesis

Descriptive statistics (mean, standard deviation, percentages) were used to synthesize the characteristics and findings of each study. Proportion was used for categorical data, while mean was used for continuous data.

The meta-analysis was performed using “Review Manager 5.4” (The Nordic Cochrane Center, https://training.cochrane.org/online-learning/core-software-cochrane-reviews/revman; accessed on 4 August 2022). The meta-analysis was performed only when at minimum two RCTs—comparable in terms of PICO parameters—investigated at least one of the defined outcomes. Due to the wide methodological heterogeneity of the included trials, we considered standardized mean difference (SMD) with 95% CI, using a random-effects model. An effect size ranging from 0.2 to 0.49 is considered “small,” from 0.5 to 0.79 is “moderate”, and if greater than 0.8, it is considered “large”. Heterogeneity was measured with the “I^2^ statistic”. The interpretation of “I^2^ values” was as follows: 0–40% “no importance”, range 30–60% “moderate”. range 50–90% “substantial”, and 75% or above “considerable” [22]. The overall quality of evidence was established using the “Grading of Recommendations Assessment, Development, and Evaluation” (GRADE) criteria. Such a framework considers five key domains (risk of bias, inconsistency, indirectness, imprecision, and publication bias), allowing reviewers to downgrade evidence from “high” to “very low” [23].

## 3. Results

### 3.1. Studies Selection

The research strategy found 2371 records. After removing the duplicates (289), 2082 articles were analyzed reading title and abstract, and 2057 were rejected since they did not meet the eligibility criteria. Among the remaining 25 studies, for one report, it was not possible to retrieve neither the abstract nor the full text, and it was not possible to contact the authors [24]; therefore, it was excluded. Twenty-four articles were assessed in the full text for eligibility. Fifteen studies did not meet the inclusion criteria and were excluded with reason (Figure 1). Finally, nine studies were included in the systematic review (Figure 1). The total number of subjects in the included studies is 1368, even if great differences were detected, with the sample size ranging from 28 to 720.

### 3.2. Description of the Studies

Seven (78%) of the included studies (n = 9) were RCTs with a parallel design [4,25,26,27,28,29,30], one was a retrospective cohort study [13], and one a retrospective case–control study [31].

All the studies had no active treatment as a comparison: three trials (33%) provided the same osteopathic evaluation, which was also delivered to the OMT group [25,26,27], and five studies (56%) provided standard medical care and/or parents recommendations [13,28,29,30,31], while one trial (11%) performed sham OMT [4].

The total number of participants was 1368, with a mean of 152 ± 215.79. Five studies (56%) involved term infants [4,26,30,31,32]; data about age were reported with different measures; therefore, it was not possible to calculate a total mean for age in term infants. Then, three studies (33%) involved preterm infants with a mean gestational age (weeks) of 33.33 ± 1.47 [13,26,27]; only one trial (11%) was conducted on very preterm infants (median gestational age in days: 187.5) [29]. Further details are shown in Table 1.

Two studies (22%) had LOS reduction as the primary outcome [26,27], two (22%) colic crying [25,29], one complete meconium excretion (11%) [25], one breast feeding at 1 month (11%) [30], one time of oral feeding (11%) [13], one infant’s biomechanical sucking difficulties [4], and one general health status (11%)—including vomiting, food intolerance, colic suggested, diarrhea—during the first 6 months of life [31].

Secondary outcomes included feeding amount and full enteral feeding [28], weight gain [13,26,27,28,31] maternal feeding perception [4], colic severity [29], costs for hospitalization [26,27], and AEs related to OMT [13,26,27,28,29,30,31].

Concerning the intervention in the experimental group, there is a consistent variability among the studies for type of techniques, number of sessions, and frequency: two RCTs evaluated the effectiveness of craniosacral therapy (CST) (22%) [25,26,27,28,29], one study provided a standardized OMT algorithm (11%) [28], and the others (67%) performed different osteopathic techniques (myofascial release, balanced ligamentous/membranous tension, indirect fluidic and v-spread, visceral treatment, cranial sutures, articulation, muscle and bone treatment) depending on the structures connected to the dysfunctional areas [4,13,26,27,30,31]. The number of OMT sessions was 1–2 per week, and only one trial performed three OMT sessions during the first week of life [28]. The duration of each session (osteopathic evaluation + OMT) varied from a minimum of 20 to a maximum of 60 min (30 ± 16.73). Further details are shown in Table 1.

### 3.3. Outcomes

All the included studies evaluated the gastrointestinal function either as a primary or secondary outcome; due to this wide inclusion criterion, there is heterogeneity among the considered outcomes and their assessment tools.

As regards infantile colic, two studies used self-reported total hours of crying and sleeping per day [25,29]; moreover, Castejòn-Castejòn et al. [29] also assessed colic severity via the infant colic severity questionnaire (ICSQ). Mills et al. [31] considered infantile colic, but a more detailed description of the outcome measurement is not provided.

Breastfeeding was assessed in two studies using different outcomes: Jouhier et al. [30], with exclusive breastfeeding at 1 month as the primary outcome and using the infant breastfeeding assessment tool (IBFAT) among the secondary outcomes, and Herzaft-Le Roy et al. [4], with the LATCH assessment tool as the primary outcome and evaluating maternal perceptions via the visual analog scale (VAS) and questionnaires as secondary outcomes.

Time to oral feeding was assessed in two studies [13,28] which also evaluated length of stay together with Cerritelli et al. [26,27]. Moreover, Haiden et al. [28] measured the complete meconium excretion as the primary outcome.

Last but not least, body weight was assessed in four studies with preterm infants [13,26,27,28] and in one study with term infants [30]. Further details are shown in Table 1.

### 3.4. Risk of Bias

Risk of bias (RoB) was assessed in RCTs [4,25,26,27,28,29,30], while ROBINS-I was assessed in observational studies [13,31].

In RoB2, five out of seven studies (71%) were judged to be at low RoB for the randomization process (Domain 1) [4,26,27,29,30], while one study was classified as some concerns, since it did not provide any information about the allocation concealment [25], and one as high RoB, since it openly stated that the details of randomization were known by the investigators and the site staff [28]. Two studies (22%) were classified as some concerns for deviations from the intended intervention (Domain 2), [25,29], while the remaining RCTs (77%) had low RoB [4,26,27,28,29]. All the assessed trials had no significant missing outcome data (Domain 3). Measurement of the outcome (Domain 4) revealed that only two trials were with some concerns, because the outcome assessors were probably aware of the group assignment [25,29]. Eventually, two out of seven studies were judged as some concerns for selection and reported results (Domain 5) since the reviewers could not find any report of pre-specified analysis in protocols or trial registries’ records such as ClinicalTrials.gov [4,25]. Therefore, the overall RoB judgment revealed that three RCTs had low [26,27,30], three had some concerns [4,25,29], and one had high risk of bias [28]. Results of RoB2 assessment and judgment for each trial are summarized in Figure 2 and Figure 3.

The two studies analyzed with the ROBINS-I showed low risk of bias in all domains, with an overall low risk of bias [13,31].

### 3.5. Description of Results

The three studies assessing infantile colic highlighted a statistically significant and progressive reduction across time in colic crying hours in infants who received OMT [25,29,30]. Hayden, Mullinger [25] showed a mean reduction in crying time of 1.0 with *p* < 0.02 in the between-group difference, Mills et al. [31] had *p* = 0.04, and Castejòn-Castejòn et al. [29] had a between-group difference of –2.47 h, *p* < 0.0005, d = 1.73 on Day 7, −3.29 h, *p* < 0.0005 d = 2.87 on Day 14, and −3.20 h, *p* < 0.0005, d = 2.54 on Day 24; moreover, Castejòn-Castejòn et al. [29] had statistically significant results in favor of OMT also regarding colic severity (−12.08 points, *p* < 0.0005, d = 1.82 on Day 7; −17.31 points, *p* < 0.0005, d = 3.07 on Day 14; −18.55 points, *p* < 0.0005, d = 3.35 on Day 24), while Mills et al. [31] also assessed spitting/vomiting at 5 months, which was statistically significant (*p* = 0.003).

Concerning feeding in preterm infants, Vismara et al. [13] showed a statistically significant difference for reaching full oral feeding in favor of the OMT group (−5.00 days, *p* = 0.042), especially in very low birth weight infants (−7.70 days *p* = 0.026); on the other hand, Haiden et al. [28] had statistically significant results in favor of CG for the enteral feeding (median: 26 days, *p* = 0.02).

For weight gain, there was no statistically significant result in none of the five studies evaluating this outcome [13,26,27,28,30]; however, in Cerritelli et al. [26], there was a statistically significant association between birth weight and the average daily weight gain (β = −0.018, *p* < 0.001 and between milk volume at study enrollment (mL) and average daily weight gain (β = 0.059, *p* < 0.001).

Concerning breastfeeding, Herzaft-Le Roy et al. [4] showed a statistically significant improvement for the infants’ ability to latch in favor of OMT (mean score: 9.22, *p* = 0.001) and in maternal perceptions concerning feeding (*p* < 0.05); on the other hand, Jouhier et al. [30] did not find any statistically significant difference neither for breast milk feeding (*p* > 0.05) nor for IBFAT (*p* = 0.3).

Only one trial assessed meconium excretion, and the results were not statistically significant (first meconium excretion: *p* = 0.16; last meconium excretion: *p* = 0.11) [28]; the same trial also assessed feeding amount measured at 14th day of life, but with no statistically significant results (*p* = 0.74). Further details are shown in Table 2.

Four studies measured LOS: in Cerritelli et al. [26] and Cerritelli et al. [27], there was a statistically significant result in favor of OMT (mean LOS average 26.1, *p* < 0.03 and 13.8, *p* < 0.001, respectively), while in Haiden et al. [28] and Vismara et al. [13], there was no difference in the between-group comparison (*p* > 0.05).

Eventually, seven out of nine studies assessed safety and OMT was substantially safe [13,28,29,30], as a matter of fact, only Haiden et al. [28] reported a transient AE with one infant showing agitations and signs of discomfort, which disappeared after a 5 min break. Instead, Herzaft-Le Roy et al. [4] and Hayden, Mullinger [25] did not evaluate if AEs occurred.

### 3.6. Effect of Interventions: Quantitative Synthesis

Two out of nine studies were included in the meta-analysis considering the outcome of “hours of crying due to infantile colic” (overall sample size: 82) [25,29]. The other studies were excluded from the meta-analysis for different reasons: two studies were not RCTs [13,31], and the other studies did not investigate infantile colic [4,13,26,27,28,30,31]. Moreover, it was not possible to conduct a meta-analysis for the other gastrointestinal outcomes because either the studies differed too consistently regarding the outcome measures, or they did not provide enough data.

Both trials [25,29] showed statistically significant effects in favor of OMT for the reduction of crying hours per day due to infantile colic (ES = −2.46 [−3.05, −1.87]; *p* < 0.00001), suggesting that OMT should be considered superior compared to no intervention (see Figure 4). Heterogeneity was low (I^2^ = 0%; *p* = 0.81). The level of evidence was rated as “moderate” (see Table 3 for further details).

## 4. Discussion

To our knowledge, this is the first systematic review and meta-analysis specifically focused on OMT and its effects on gastrointestinal disorders in preterm and term infants. In fact, a Cochrane meta-analysis [15] and a previous systematic review [16] analyzed the effects of manual therapy—thus including not only OMT as the experimental intervention—and the pediatric conditions in general, respectively; moreover, they were published in 2012, thus requiring an update in order to include more recent studies.

In general, the included studies had conflicting evidence on the effectiveness of OMT for gastrointestinal disorders, although OMT has been shown to be a safe intervention and provides new clinical and scientific insights into the treatment of gastrointestinal disorders in the pediatric sector. In fact, Mills et al. [31] and two RCTs [25,29] showed statistically significant results for infantile colic treated with OMT: *p* = 0.04, *p* < 0.02, and *p* < 0.0005, respectively; moreover, the two RCTs [25,29] had an overall effect size in favor to OMT (ES = −2.46 [−3.05, −1.87]; *p* < 0.00001). However, they both presented a high risk of bias in the overall RoB grade, thus meaning that the results have to be interpreted carefully. Other two trials analyzed breastfeeding, and they had opposite results [4,30]: in the study by Herzhaft-Le Roy et al. [4] there was a statistically significant improvement for latching and maternal perceptions on feeding (mean score: 9.22, *p* = 0.001), but it was judged with some concerns in the overall RoB grade; on the other hand, Jouhier et al. [30] had a methodologically stronger study—with low RoB in the overall grade—but nonstatistically significant results (*p* > 0.05). Instead, for the preterm infants’ feeding Vismara et al. [13] showed statistically significant reduction of time to full oral feeding exposed to OMT (*p* = 0.042), while Haiden et al. [28] had statistically significant results for the full enteral feeding in the control group (*p* = 0.02). Eventually, all the five studies [13,26,27,28,30] assessing birth weight did not show any statistically significant improvement (*p* > 0.05); however, birth weight was not the primary outcome in these studies; therefore, it is possible that they were not powered enough to find a difference.

Undoubtedly, there are some limitations to these studies that prevent generalizing the results, such as the variability of the interventions, which differed in techniques, length of the sessions, frequency, and number of treatments. However, it is important to note that this heterogeneity is common in OMT trials, and it cannot be prevented since it is intrinsic and directly related to the nature of OMT itself; as a matter of fact, OMT is a patient-centered approach with many techniques that are performed depending on the patients’ needs, thus preventing a standardized protocol of intervention [32]. Moreover, another aspect that prevents generalizing the results and is important to take into account is the paucity of the studies; as a matter of fact, just in a few cases, the studies evaluated the same outcome, and—when it occurred—the measurement tools could differ.

The osteopathic assessment of the somatic dysfunction followed by osteopathic techniques can positively influence the fascial system and consequently the autonomic nervous system favoring the parasympathetic response, which is strictly related to the gastrointestinal system [10,14,33]; therefore, enhancing the infants’ adaptation, specifically regarding the fascial and autonomic nervous system, could explain the improvement for the gastrointestinal function. Thanks to the bidirectional communication network between the gut and the nervous system, the GT is strictly connected with emotion and stress. For instance, focusing on preterm infants, it has been shown that maternal separation in the NICU procedures—which are often invasive—causes alterations in the functions of the GT. In fact, the stress switches on the hypothalamic–pituitary–adrenal axis and the sympathetic nervous system, subsequently leading to gut permeability, allowing bacteria and bacterial antigens to activate the mucosal immune response, thus altering the microbiota [34]. Moreover, considering feeding, the earlier start of the full oral feeding and the improvement of the sucking capacities can also bring to better microbiome; in fact, it has been shown that microbial communities differ between breastfed and formula-fed infants: breastfed infants have different bacterial classes and a higher interaction, subsequently having a positive impact on the immune response and the metabolic activities [35]. Considering the importance of microbiota, the effects of the OMT on the autonomic nervous system, and this bidirectional communication, it would be interesting to assess the effects of OMT when combined with probiotics; as no studies combining these two interventions were retrieved, it seems that this is still an unknown pathway.

Despite the increased research on OMT, there is still a need for high-quality studies in the pediatric field to make a proper comparison and fully understand the role of OMT in the current panorama. This is a necessity which was also stated in previous pediatric reviews evaluating osteopathy; in fact, a Cochrane systematic review and meta-analysis assessing the effects of manual therapies on infantile colic—thus also including OMT trials—mentioned that the little number of studies and the small sample size prevent generalization [15]. Then, there is also a systematic review evaluating the effects of OMT in the pediatric population in general, without focusing on a specific system or condition, which underlined the lack of replication studies [16].

### 4.1. Quality of Evidence

The quality of evidence (see Table 3) was judged as “moderate”.

First of all, a downgrade was performed due to the relevant risk of bias, mostly related to blinding procedures, deviation from the intended intervention, and measurement of the outcome. Then, we downgraded for imprecision due to the width of the confidence intervals. Instead, we did not downgrade for inconsistency because of the low heterogeneity coming from all the analyses (I^2^: 0%). As known, a “moderate” quality of evidence implies that the true effect is likely to be close to the estimate of effect, even if—at the same time—there is a possibility that it is substantially different from the real estimate of the effect. These findings should encourage further high-quality studies, attempting to increase and to improve research in the osteopathic field, with the aim of raising the overall quality of evidence. This could strengthen recommendations for clinicians, researchers, and healthcare policies.

### 4.2. Strengths and Limitations

It was not possible to include all the studies in the meta-analysis due to the high variability among the outcomes and the limited number of studies; moreover, even if it was possible to calculate the overall effect of the hours of crying due to infantile colic, we could include only two studies, highlighting the high necessity to increase research projects in this field. Then, another limit is publication bias, which was not assessed since no statistical tool is definitely able to detect it [36], and therefore, it may be present.

Despite its limits, this systematic review and meta-analysis is highly innovative since it is the first study assessing the effects of OMT on gastrointestinal disorders in term and preterm infants. In fact, the summary of the different results and the risk of bias assessment describe the current evidence, thus providing useful insights for future research.

## 5. Conclusions

Concerning gastrointestinal disorders in term and preterm infants, OMT is overall safe and provides many clinical and research insights. In terms of safety, no included study reported serious sequelae even when treating a sensitive population such as preterm infants. Among AEs, only one study reported a mild one, which resolved spontaneously in a short time [28]. To date, OMT effectiveness has not been shown yet due to the lack of high-quality and replication studies; further RCTs with OMT as an add-on therapy compared to the usual care alone can help to clarify its support in clinical practice, and possibly provide material to establish future guidelines accordingly. OMT may be a new field of clinical study and treatment in GI disorders in the pediatric sector.

## Figures and Tables

**Figure 1 healthcare-10-01525-f001:**
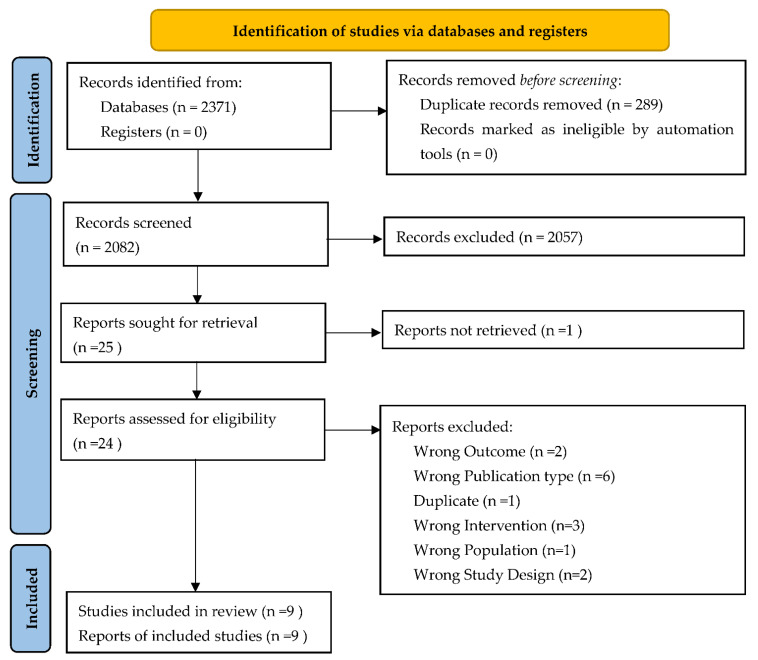
Flow diagram based on PRISMA statement.

**Figure 2 healthcare-10-01525-f002:**
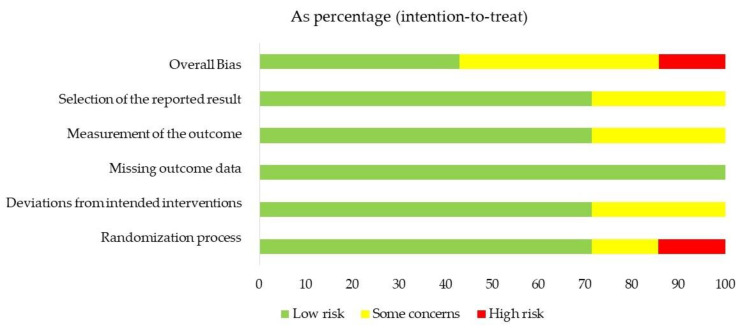
Risk of bias assessment graph of the included studies.

**Figure 3 healthcare-10-01525-f003:**
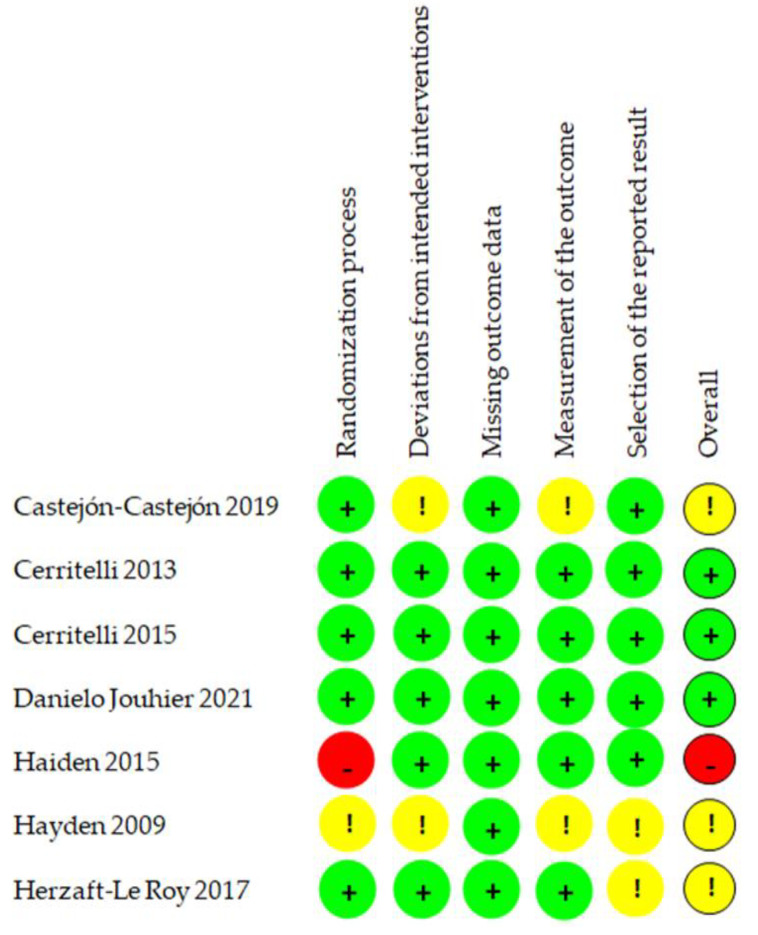
Risk of bias assessment graph of the included studies.

**Figure 4 healthcare-10-01525-f004:**
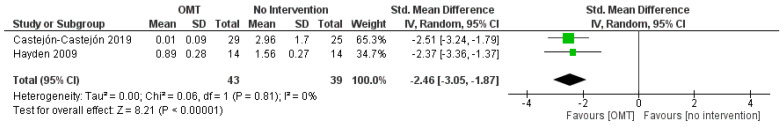
Forest plot comparing the effects of OMT vs. no intervention for hours of crying per day due to infantile colic in term infants.

**Table 1 healthcare-10-01525-t001:** Overview of included studies.

Author Year	Study Design	Objective	Outcomes/ Variables	Population	Intervention	Comparison
**Hayden et al.** **[25] 2009**	RCT	Efficacy of CST on colic in infants between 1 and 12 weeks of age	(1) Hours/24 h spent with colicky crying * (2) Number of hours/24 h spent sleeping *	N = 28 Male: 79% Age at study entry (days): 45.45 ± 5.2 GA at delivery (days): 276 ± 2.35	OMT (n = 14) *Description*: 5 sessions of CST for 4 weeks (first session with 60 min duration; other sessions with 30 min duration)	CG (n = 14) *Description*: no intervention
**Cerritelli et al.** **[26] 2013**	RCT	Efficacy of OMT on LOS in preterm infants	(1) LOS * (2) Daily weight gain (3) Cost reduction	N = 110 Male: 49% GA (weeks): 34 ± 2.4 Days of life: 3.4 ± 2.4 BW (g): 2161 ± 614.75	OMT (n = 55) *Description*: OMT (20 min duration twice per week) plus usual care	CG (n = 55) *Description*: osteopathic evaluation (10 min of evaluation + 10 min without touch twice per week) plus usual care.
**Cerritelli et al.** **[27] 2015**	RCT	Efficacy of OMT on LOS in preterm infants	(1) LOS * (2) Daily weight gain (3) Cost reduction	N = 720 Male: 50% GA (weeks): 34.35 ± 2.25 Days of life: 3.65 ± 2.25 BW (g): 2299.5 ± 731.15	OMT (n = 352) *Description*: OMT sessions twice a week until discharge plus usual care. OMT sessions lasted 30 min (10 min for evaluation and 20 min for treatment)	CG (n = 343) *Description*: structural evaluation (10 min of evaluation + 20 min without touch) plus usual care
**Haiden et al.** **[28] 2015**	RCT	Efficacy of OMT on meconium passage in very low birth weight preterm infants	(1) Complete meconium excretion * (2) Introduction of enteral feeding in days (3) Feeding volume on Day 14th (4) Time to full enteral feeding in days (5) LOS (6) Weight at discharge	N = 41 GA (days): 187.5 (165–211) BW (g): 747.5 (441.5–1275)	OMT (n = 21) *Description*: standardized OMT algorithm within the first 48 h of life and on 3 days during the first week of life	CG (n = 20) *Description*: no intervention (only standard medical care)
**Herzaft-Le Roy et al.** **[4] 2017**	RCT	Efficacy of OMT combined with lactation consultations on infants’ biomechanical sucking difficulties	(1) LATCH * (2) Mothers’ nipple pain (3) Questionnaire about maternal perceptions (4) Questionnaire about side effects (5) Questionnaire about breastfeeding management	N = 97 Male: 47% Age at study entry (days): 15 ± 10.41	OMT (n = 49) *Description*: one session of OMT (30 min duration) plus two lactation consultations	CG (n = 48) *Description*: one sham manipulation (30 min duration) plus two lactation consultations
**Castejón-Castejón** **et al. [29] 2019**	RCT	Effectiveness of OMT on colic in infants	(1) Total hours of crying per day * (2) Sleep (3) Colic severity (ICSQ)	N = 58 Male: 50% GA (weeks): 36.41 ± 18	OMT (n = 29) *Description*: from one to three sessions of CST at one week distance (the number of sessions depended on the presence or not of colic)	CG (n = 29) *Description*: no intervention
**Vismara et al.** **[13] 2019**	Retrospective cohort study	Effects of OMT on TOF in very/moderately preterm infants	(1) TOF (2) Body weight (3) Body length (4) Head circumference (5) LOS	N = 70 Male: 50% GA (weeks): 31.65 ± 1.7 BW (g): 1483.75 ± 281.7	OMT (n = 35) *Description*: two OMT sessions per week since the first two weeks of life (30 min duration) plus usual care	CG (n = 35) *Description*: usual care
**Danielo Jouhier et al.** **[30] 2021**	RCT	Efficacy of OMT on breast feeding at 1 month	(1) Exclusive breast milk feeding at 1 month * (2) Exclusive breast milk feeding at 3 months (3) IBFAT at 10 days (4) Infant’s body weight at 10 days (5) Infant’s body weight at 1 month (6) Maternal satisfaction at 10 days (7) Maternal satisfaction at 1 month (8) Acute neonatal pain scale score (9) AEs linked to OMT	N = 128 Male: 52% GA (weeks): 39.70 BW (g): 3466.5 ± 348	OMT (n = 59) *Description*: 2 sessions of OMT (the first session before discharge and the second at one week distance)	CG (n = 59) *Description*: no intervention (manipulation on a doll placed next to the infant)
**Mills et al.** **[31] 2021**	Case–control study	Effects of OMT on health in the first 6 months of life	(1) Spitting/vomiting (2) Described as gassy (3) Food intolerance (4) Irritability/Sleep (5) Colic suggested (6) Otitis media (7) Antibiotics given (8) Upper respiratory infections (9) Lower respiratory problems (10) Diarrhea (11) Rashes	N = 116 Male: 53%	OMT (n = 58) *Description*: one or two sessions, depending on the infant’s LOS (5-10 min duration)	CG (n = 58) *Description*: no intervention

*: Primary outcome. N: sample size. Abbreviations. RCT: randomized controlled trial; CST: craniosacral therapy; GA: gestational age; OMT: osteopathic manipulative treatment; CG: control group; LOS: length of stay; BW: birth weight; LATCH: latch, audible swallowing, type of the nipple at the end of the feed, comfort, and how mother is able to hold her infant to the breast; ICSQ: infant colic severity questionnaire; TOF: time to oral feeding; AEs: adverse events; IBFAT: infant breastfeeding assessment tool. *Note.* All the continuous variables are expressed with mean and standard deviation, except for Haiden et al. [28], where they are summarized with median and range.

**Table 2 healthcare-10-01525-t002:** Description of interventions and main results of the included studies.

Author Year	Study Design	Description of Interventions	Main Results
**Hayden et al. [25]** **2009**	RCT	OMT: CST After an evaluation with minimal touch, the techniques were performed until a palpable release of tensions and dysfunction was achieved. CG: no intervention. Same examination of the OMT group. 5 sessions for 4 weeks (initial visit: 60 min; following visits: 30 min).	*Colic crying:* There was a statistically significant difference between OMT and CG for the mean hours of colic crying in hours/24 h in favor of the OMT group (*p* < 0.02). *AEs:* None.
**Cerritelli et al.** **[26] 2013**	RCT	OMT: standard medical care + OMT (myofascial release, balanced ligamentous/membranous tension, indirect fluidic and v-spread). CG: standard medical care + osteopathic evaluation. 2 sessions per week, lasting 20 min (10 min for the evaluation and 10 min for the treatment for the OMT group; 10 min for the evaluation and 10 with the osteopath standing in front of the incubator in the control group).	*Weight gain:* There was no statistically significant association between OMT and the average daily weight gain (*p* = 0.06); instead, there was a statistically significant association between birth weight and the average daily weight gain (*p* < 0.001) and between milk volume at study enrollment (mL) and average daily weight gain (*p* < 0.001). *LOS reduction:* There was no statistically significant reduction in LOS in the OMT group compared to CG (*p* < 0.03). *AEs:* None.
**Cerritelli et al. [27]** **2015**	RCT	OMT: standard medical care + OMT (myofascial release and balanced ligamentous/membranous tension). CG: standard medical care + osteopathic evaluation. The sessions occurred twice per week, lasting 30 min (10 min for the evaluation and 20 min for the treatment for the OMT group and 10 min for the evaluation, and 20 with the osteopath standing in front of the incubator in the control group).	*Weight gain:* There was no statistically significant association between OMT and the average daily weight gain (*p* = 0.35); instead, there was a statistically significant association between birth weight and the average daily weight gain (*p* < 0.01). *LOS reduction:* There was a statistically significant reduction of 3.9 days in LOS in the OMT group compared to CG (*p* < 0.01). *AEs:* None.
**Haiden et al. [28]** **2015**	RCT	OMT: standard medical care + OMT algorithm (global and local listening of the abdomen, release lower ribs and thoracic diaphragm, pylorus relaxation, release of the duodenum and the C-loop, small intestine diagnosis—lifting the gut and bringing it to a stillpoint, mobilization of the ileocecal valve, mobilization of colon ascendens, transversum and descendens with treatment of the Toldt fascia, root of sigmoid diagnosis and manipulation, treatment of the vagus nerve with CST via the sacrum). CG: standard medical care. There were a total of 3 sessions during the first week of life, and the OMT algorithm was repeated 3 times during each session.	*Meconium excretion:* There was no statistically significant difference between OMT and CG for the first meconium excretion (*p* = 0.16) and the last meconium excretion (*p* = 0.11). *Feeding amount on the 14th day of life:* There was no statistically significant difference between OMT and CG for the feeding amount on the 14th day of life (*p* = 0.74). *Full enteral feeding:* There was a statistically significant difference between OMT and CG for the full enteral feeding in favor of CG (*p* = 0.02); in fact, time to full enteral feedings was 8 days longer in the intervention group (median 34 days, 95% Cl: 30–48 days) than in the control group (median 26 days, 95% Cl: 20–31 days). *Weight gain:* There was no statistically significant difference between OMT and CG for the weight at discharge (*p* = 0.58). *LOS:* No statistically significant difference between OMT and CG for LOS (*p* = 0.14). *AEs:* No infant showed signs of cardiorespiratory instability, apnea, or pain. Only 1 infant (4.8%) showed agitation and signs of discomfort; however, after a 5 min break, the infant calmed down, and the treatment was continued without further problems.
**Herzaft-Le Roy** **et al. [4] 2017**	RCT	OMT: lactation consultation (emotional support and better positioning of mothers and babies) + OMT (balanced membranous tension, cranial sutures, and myofascial release). CG: lactation consultation (emotional support and better positioning of mothers and babies) + sham OMT (light touch far from the osteopathic dysfunctional areas found). The OMT and sham OMT sessions lasted 30 min, while the lactation consultations lasted 60 min.	*Latching:* There was a statistically significant difference between OMT and CG for the infants’ ability to latch measured with the LATCH score in favor of the OMT group at Day 3 (*p* = 0.001). *Maternal perceptions concerning feeding:* There were statistically significant differences between OMT and CG regarding the infants’ ability to open the mouth widely (*p* < 0.016), nipple biting (*p* < 0.042), and the tendency for the infants’ mouth to slip on the nipple (*p* < 0.002) in favor of the OMT group at Day 3. *AEs:* None.
**Castejón-Castejón** **et al. [29] 2019**	RCT	OMT: CST (balance of the pelvic and thoracic and clavicular diaphragms) + written recommendations on how to take care of a baby with infantile colic. CG: no intervention (only written recommendations on how to take care of a baby with infantile colic, the same provided to the OMT group). Infants in the OMT group received 1 to 3 CST sessions (depending on the presence of colic symptoms) lasting 30–40. The sessions occurred at Day 1 (baseline) and—when required—at Day 7 and Day 14.	*Crying:* There was a statistically significant difference between OMT and CG for the crying hours in favor of the OMT group at Day 7 (*p* < 0.0005; d = 1.73), Day 14 (*p* < 0.0005; d = 2.87) and Day 24 (*p* < 0.0005; d = 2.54). Moreover, the rANCOVA considering the respective baseline values as covariates showed statistically significant results (*p* = 0.000). *Colic severity:* There was a statistically significant difference between OMT and CG for colic severity in favor of the OMT group at Day 7 (*p* < 0.0005; d = 1.82), Day 14 (*p* < 0.0005; d = 3.07) and Day 24 (*p* < 0.0005; d = 3.35). *AEs:* None.
**Vismara et al. [13]** **2019**	Retrospective cohort study	OMT: standard medical care + OMT (treatment of the myofascial and connective tissues). Treated areas: cranial (cranial techniques) and occipital, the C1-C2-C3 areas, hyoid, sacrum, diaphragm, upper chest, scapulae, left iliac fossa and the structures connected in anatomical and physiological ways to these structures. CG: standard medical care. Sessions started in the first 2 weeks of life with a frequency of twice per week, lasting at least 30 min.	*Time to oral feeding:* There was a statistically significant difference between OMT and CG for the full oral feeding in favor of the OMT group (*p* = 0.042). Moreover, a post-hoc analysis with a stratification by body weight at birth showed a statistically significant difference between OMT and CG for the full oral feeding in VLBW infants in favor of the OMT group (*p* = 0.026); instead, no statistically significant difference was found in LBW infants (*p* = 0.096). *Weight gain:* There was no statistically significant difference between OMT and CG for body weight (*p* = 0.672). *LOS:* There was no statistically significant difference between OMT and CG for LOS (*p* = 0.065) *AEs:* None.
**Danielo Jouhier** **et al. [30] 2021**	RCT	OMT: CST, muscular, bones, and/or visceral treatment depending on the found dysfunctional areas. CG: no intervention. The osteopath manipulated a doll placed next to the infant in order to avoid revealing to the mother that the child was not being treated. 2 sessions (the first before discharge and the second 7 days later).	*Exclusive breast milk feeding:* There was no statistically significant difference between OMT and CG for breast milk feeding neither at 1 month (*p* = 0.14) nor at 3 (*p* = 0.55) and 6 months (*p* = 0.92) in the per-protocol analysis. Moreover, there was no statistically significant difference at 1 month with the intention to treat analysis neither by imputing missing data as a breastfeeding success (*p* = 0.13) nor by imputing missing data as a breastfeeding failure (*p* = 0.15). *Infant breastfeeding assessment tool (IBFAT):* There was no statistically significant difference between OMT and CG for IBFAT at Day 10 (*p* = 0.3). *Weight gain:* There was no statistically significant difference between OMT and CG for weight gain at 1 month (*p* = 0.9). *AEs:* None.
**Mills et al. [31]** **2021**	Case–control study	OMT: standard medical care + OMT (articulation, direct and indirect myofascial release, balanced membranous tension, and balanced ligamentous tension). CG: standard medical care. 1 or 2 sessions depending on the length of the baby’s hospital stay, lasting 5–10 min.	*Spitting/vomiting:* There was a statistically significant difference between OMT and CG for spitting/vomiting at month 5 (*p* = 0.003). *Colic suggested:* There was a statistically significant difference between OMT and CG for colic suggested at month 3 (*p* = 0.04). *AEs:* None.

*p*: *p*-value (significance level). d: Cohen’s d (effect size). Abbreviations. OMT: osteopathic manipulative treatment; CST: craniosacral therapy; CG: control group; AEs: adverse events; IBFAT: infant breastfeeding assessment tool; VLBW: very low birth weight; LBW: low birth weight.

**Table 3 healthcare-10-01525-t003:** Quality of evidence assessed through GRADE framework.

Outcome	SMD (95% CI)	N. of Subjects (Studies)	Comments	Quality of Evidence
Hours of crying per day (infantile colic)	−2.46 (−3.05, −1.87)	82 (2 studies)	Downgraded by 1 level for RoB Downgraded by 1 level for Imprecision	⊕⊕◯◯

**High quality**: We are very confident that the true effect lies close to that of the estimate of the effect. **Moderate quality**: We are moderately confident in the effect estimate; the true effect is likely to be close to the estimate of effect, but there is a possibility that it is substantially different. **Low quality**: Our confidence in the effect estimate is limited; the true effect may be substantially different from the estimate of the effect. **Very low quality**: We have very little confidence in the effect estimate; the true effect is likely to be substantially different from the estimate of the effect.

## Data Availability

Not applicable.

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
