# Peer review of "Osteopathic Treatment for Gastrointestinal Disorders in Term and Preterm Infants: A Systematic Review and Meta-Analysis"

_healthcare, 2022, doi:10.3390/healthcare10081525_

Round 1

Reviewer 1 Report

The authors report on a systematic review of Osteopathic treatment for gastrointestinal disorders in term and preterm infants.

After reading the manuscript I have some concerns that should be resolved:

What was the difference between your study from other reviews?

You should emphasize the effects of OMT on the GT in the introduction section.

More information about the search strategy as a supplementary file would be helpful.

Authors should check the Flow diagram. The numbers are wrong… 2082-2057=25…. Otherwise, you should include 10 studies.

Mean session time can be given in the result section

Please provide a summary of the results in the first paragraph of the discussion

Most importantly, more in-depth discussion is needed to strengthen the impact of this study.

Author Response

REVIEWER 1

The authors report on a systematic review of Osteopathic treatment for gastrointestinal disorders in term and preterm infants.

Dear reviewer,

As a premise, we are grateful for the time you have devoted to our work, for your general judgement and for your useful suggestions aimed to improve the quality of the paper. We tried to acknowledge them point-by-point; we highlighted all the changes in the revised paper.

Thank you.

After reading the manuscript I have some concerns that should be resolved:

What was the difference between your study from other reviews?

Thank you for addressing this question; we outlined the significance and innovation of our review in the first paragraph of the Discussion (page 7)

You should emphasize the effects of OMT on the GT in the introduction section.

Thank you for highlighting this point; we added the explanation at page 2.

More information about the search strategy as a supplementary file would be helpful.

We included the search strategy used in PubMed. Since for the other databases we used the same strategy, we did not insert it, otherwise it would have been a repetition in the document. However, we included the missing specification in the text (page 2).

Authors should check the Flow diagram. The numbers are wrong… 2082-2057=25…. Otherwise, you should include 10 studies.

Thank you for noticing this inconsistency, we controlled and modified the flowchart (see figure 1 and page 4).

Mean session time can be given in the result section

Thank you for this suggestion; we included the mean and standard deviation for session time at page 5.

Please provide a summary of the results in the first paragraph of the discussion

Thank you for outlining this aspect; we modified the paper accordingly (page 7)

Most importantly, more in-depth discussion is needed to strengthen the impact of this study.

Thank you for this suggestion; we emphasized the novelty of the study in the Discussion (page 7) and in the Conclusions (page 8 and 9)

Reviewer 2 Report

Dear Authors,

I've read with great interest your interesting this sistematic review. Each section is very well oraginzed, with sufficient arguments to sustain the aim of the study. It provides great singificance for the community. However, I have one small request before publication. If possible, please, check again the entire manuscript for font inconsistencies since I identifed several of them. Other than that, great work.

Indeed. I identified several issues that should be corrected by the authors. 1) The authors should provide several arguments regarding the novelty of this study in contrast to the actual state of knowledge. 2) I want to know how the authors manage to remove the duplicates. 3) The authors could provide more details in the Introduction section regarding OMT on the GI 4) Unfortunately, the number of eligible studies is low, which is why the results might be relative. 5) The authors should add some info in the Discussion section regarding if there is a correlation between such approaches and newborn microflora development and possible consequences later in life. 6) The authors could state what was the main concern and objective when they referred to "gastrointestinal function" and "gastrointestinal disorder". They could detail a little bit more about the role of the gastrointestinal microbiota, particularly "digestive, absorptive, neuroendocrine and immunologic functions"? 7) Why the authors did not retrieve that one study? 8) Why the authors did not use other databases (ISI Web of Knowledge, Scopus, ScienceDirect?) 9) The authors did not find any article combining osteopathy with probiotic intervention?   In this context, a Major Revision is needed.

Kind regards and all the best,

The Reviewer

Author Response

REVIEWER 2

Dear Authors,

I've read with great interest your interesting this sistematic review. Each section is very well oraginzed, with sufficient arguments to sustain the aim of the study. It provides great singificance for the community. However, I have one small request before publication

Dear reviewer,

As a premise, we are grateful for the time you have devoted to our work, for your general judgement and for your useful suggestions aimed to improve the quality of the paper. We tried to acknowledge them point-by-point; we highlighted all the changes in the revised paper.

Thank you.

If possible, please, check again the entire manuscript for font inconsistencies since I identifed several of them. Other than that, great work.

Thank you for your suggestion. We have done what is requested.

Indeed. I identified several issues that should be corrected by the authors.

1) The authors should provide several arguments regarding the novelty of this study in contrast to the actual state of knowledge.

Thank you for your suggestion, we added the explanation at page 2 of the Introduction and 7 of the Discussion.

2) I want to know how the authors manage to remove the duplicates.

Thank you for highlighting this step of the study selection. According to your suggestion, we added in the text (page 3) how we removed the duplicates.

3) The authors could provide more details in the Introduction section regarding OMT on the GI

Thank you for your suggestion, we added the explanation at page 2.

4) Unfortunately, the number of eligible studies is low, which is why the results might be relative.

It is absolutely true, however, they are clinically very significant providing an accurate analysis on the studied topic.

5) The authors should add some info in the Discussion section regarding if there is a correlation between such approaches and newborn microflora development and possible consequences later in life.

Thank you for suggesting to expand this topic. We added this information (page 8)

6) The authors could state what was the main concern and objective when they referred to "gastrointestinal function" and "gastrointestinal disorder". They could detail a little bit more about the role of the gastrointestinal microbiota, particularly "digestive, absorptive, neuroendocrine and immunologic functions"?

Thank you for suggesting to expand this topic. We added this information (page 8)

7) Why the authors did not retrieve that one study?

We understood that it was important to explain why the study could not be retrieved, therefore we added the explanation in the text (page 4). Moreover, we also specified the study (reference 22).

8) Why the authors did not use other databases (ISI Web of Knowledge, Scopus, ScienceDirect?)

Thank you for addressing this point regarding databases. As you can see, Scopus has already been included in the first version of the manuscript. Regarding Science Direct, we decided not to include it because, as reported in the website “Elsevier” (1), Scopus and Science Direct basically share the same articles. Actually, Scopus includes even more articles compared to Science Direct since it does not contain only material published by Elsevier, but also from other journals. Concerning ISI Web of Knowledge, In reality, in a first preliminary analysis, we included it, but the database did not add any new articles compared to the other databases. So we focused on those indicated. We think we included all the articles on the topic studied.

(1) Elsevier. “What Is the Difference between ScienceDirect and Scopus Data?” What Is the Difference between ScienceDirect and Scopus Data? - Data as a Service Support Center, https://service.elsevier.com/app/answers/detail/a_id/28240/supporthub/dataasaservice/p/17729/.

9) The authors did not find any article combining osteopathy with probiotic intervention?   In this context, a Major Revision is needed.

Thank you for outlining the interesting aspect of probiotics. In our research we did not find any study combining osteopathy with probiotics in our population of interest (term and preterm infants). Indeed it would be interesting to develop new trials with OMT + probiotics. We added this lack of studies in the Discussion (page 8).

Round 2

Reviewer 1 Report

All my previous concerns and questions were addressed and answered.

Author Response

Thank you very much.

Reviewer 2 Report

Dear Authors,

Congratulations on the hard work reviewing this manuscript. Its quality has been improved.

Kind regards and all the best,

The Reviewer

Author Response

Thank you very much.